

# Exercise mode and attentional networks in older adults: a cross-sectional study

Biye Wang[1,2] and Wei Guo[1,2]

[1] College of Physical Education, Yangzhou University, Yangzhou, Jiangsu, China
[2] Institute of Sports, Exercise and Brain, Yangzhou University, Yangzhou, Jiangsu, China

## ABSTRACT

**Background**. Previous studies have indicated that physical exercise enhances attentional function; however, the relationship between exercise mode and attentional networks has not been clarified for older adults (>60 years old). This study aimed to investigate the relationship between attentional networks and different exercise modes in older adults.

**Methods**. Two hundred and fifty-nine participants aged between 60 to 81 years were enrolled and classified into three groups (closed-skill group, open-skill group, or sedentary control group) using an exercise-related questionnaire. All participants completed an attention network test (ANT), which measured executive control, orienting, and alerting networks.

**Results**. The open-skill group had significantly higher executive network efficiency compared to the closed-skill ($p < 0.01$) and sedentary ($p < 0.01$) groups. The closed-skill group had significantly higher values compared to the sedentary control group ($p < 0.05$). Differences were not detected among groups for alerting and orienting networks ($p > 0.05$). The open-skill group had significantly higher values compared to the sedentary control group regarding proportion score of executive network ($p < 0.01$). In comparison, no significant differences were detected among groups for proportion scores of alerting and orienting networks.

**Conclusion**. This study extends current knowledge by demonstrating that open-skill exercises selectively enhance the executive control of attentional networks in older adults. Open-skill exercises combines physical exercise and cognitive training, potentially representing a more effective exercise mode to maintain or enhance attentional function in older adults.

## INTRODUCTION

Age-related impairments in attention, memory, processing speed, and reasoning arise during adulthood, and progress into the elder years (*Salthouse, 2016*). Attention is the basic brain function to ensure that we interact effectively with the environment by selectively focusing on relevant information over other information (*Posner & Petersen, 1990*); however, this function is particularly age-sensitive (*Gamboz, Zamarian & Cavallero, 2010*; *Waszak, Li & Hommel, 2010*). There is increasing evidence, however, that physical exercise has a positive impact on attention. For example, a recent systematic review of

Corresponding author
Wei Guo, guowei@yzu.edu.cn

44 articles (with effect sizes ranging from 0.2 to 0.8) showed that acute exercise ($n = 23$), aerobic exercise intervention ($n = 9$), and aerobic fitness ($n = 12$) might have a positive effect on sustained and selective attention (*Fernandes M. de Sousa et al., 2018*). Another recent review (*Muiños & Ballesteros, 2018*) also showed that older adults who participate in ball sports or fast-moving sports (e.g., tennis and martial arts) exhibit better visual attention compared to sedentary adults; thus, exercise mode might be related to attention function. Also, a meta-analysis of randomized control trials demonstrated a significant improvement in the attention and processing speed of older adults who participated in Tai Chi compared to sedentary controls (*Kelly et al., 2014*).

According to attentional network theory, the human attention system is divided into three networks: orienting, alerting, and executive control (*Petersen & Posner, 2012*; *Posner & Petersen, 1990*). Our basic attention function for selecting information from multiple sensory inputs is facilitated by the orienting network, which depends on a network composed of the superior parietal lobe, superior colliculus, frontal eye fields, and temporoparietal junction (*Mayer et al., 2004*). The alerting network is mediated by the parietal areas of the dorsal visual pathway and the right frontal (*Marzo et al., 2014*), and is associated with enhancing vigilance and preparedness. The executive control network is involved in monitoring and resolving conflict. It is believed that anterior cingulate cortex, the dorsolateral prefrontal cortex, and portions of the basal ganglia and the thalamus are associated with this network (*Crottaz-Herbette & Menon, 2006*). The attention network test (ANT) was developed to assess the attentional networks independently in a single test, based on attentional network theory (*Fan et al., 2002*).

ANT has been used by previous studies to show the positive effects of martial arts training (*Johnstone & Marí-Beffa, 2018*), table tennis training (*Wang, Guo & Zhou, 2016*), chronic exercise (*Pérez et al., 2014*), and acute bouts of aerobic exercise (*Chang et al., 2015*; *Huertas et al., 2011*) on the three components of attentional networks in young adults. A recent randomized control trial showed that hatha yoga had no significant effect as an exercise intervention in older adults compared to a stretching control group regarding the three attentional networks (*Gothe, Kramer & McAuley, 2017*). In contrast to open-skill exercises (e.g., table tennis and badminton), yoga is a typical closed-skill exercise that is conducted in a relatively stable, predictable and self-paced environment. Open-skill exercises are those in which exercisers are required to react in a dynamic changing, unpredictable, and external-paced environment (*McMorris, 2014*). Several studies have demonstrated additional cognitive benefits of open-skill exercises over closed-skill exercises in older adults (*Dai et al., 2013*; *Guo et al., 2016*). The additional cognitive benefits of open-skill exercises are mainly attributed to environmental enrichment and increased cognitive demand during physical exercise (*Fabel et al., 2009*). *Diamond & Ling (2016)* argued that aerobic exercise lacks cognitive components, and so produces little or no cognitive benefit (*Diamond & Ling, 2016*). The authors suggested that physical training should extend beyond simple movement to movement with thought (*Diamond, 2015*). Thus, the present study aimed to differentiate the relationship between exercise mode and attentional networks in older adults in relation to open- and closed-skill exercise.

Several studies have demonstrated that advanced aging is accompanied by declines in attention, with physical exercise delaying such declines. However, information on how different exercise modes affects the cognitive domain in the elderly remains limited. Thus, the current study aimed to extend existing knowledge by elucidating whether different exercise modes produce different benefits (i.e., improvements) on certain components of attention network function. Specifically, this study focused on how two exercise modes affect attention network function in older adults when performing ANT. Based on existing information from the published literature, we hypothesized that: (1) both open-skill and closed-skill exercisers would perform better than sedentary controls in ANT; and (2) open-skill exercisers would be selectively better than closed-skill exercisers with respect to executive control network function.

## MATERIALS & METHODS

### Ethical approval

The study was carried out ethically, and was approved by the Ethical Committee of Shanghai University of Sport (No. 2017044).

### Participants

Participants aged between 60 to 81 years were recruited from senior centers of various communities through posted advertisements. All participants were required to satisfy the following criteria: (1) right handed, (2) had normal or corrected to normal visual acuity, (3) demonstrated no dementia by scoring higher than 25 in the Mini-Mental Status Examination (MMSE), (4) had a normal body mass index (BMI was less than 30.0 and more than 18.5), and (5) reported to be free of brain injury, psychiatric, neurological disorders, and cardiovascular disease. Fifteen applicants were excluded from participation in this step. Eligible participants were categorized into three groups (open-skill, closed-skill, and sedentary) according to their exercise modes, which were assessed by the International Physical Activity Questionnaire (IPAQ, Chinese Version) and an exercise mode questionnaire.

The Chinese Version of IPAQ has been proven to have good reliability and validity (*Liou et al., 2009*), and the exercise mode questionnaire was used to obtain details of exercise, including specific exercise mode, frequency per week, duration (per session), and the number of years participating in certain modes of exercise. The two questionaries were adopted in previous studies (*Dai et al., 2013*; *Huang et al., 2014*; *Guo et al., 2016*). The open-skill group satisfied the criteria of participating in open-skill exercise (i.e., table tennis, badminton) at least three times per week for 30 min each time in the previous year. The closed-skill group participated in closed-skill exercise (i.e., jogging, swimming) with the same frequency. Participants who took part in both open-skill and closed-skill exercises were excluded. The sedentary group reported inactivity or low activity levels in IPAQ, and no regular exercise.

According to the results of previous meta-analyses (*Northey et al., 2018*; *Young et al., 2015*), the effect size of physical exercise on cognitive function is about 0.2. A power analysis based on the effect size was conducted. The result showed that a minimum of 246 subjects

would be needed to reach a power of 0.80 to detect the effect size of 0.2. Significance was set to 0.05. In total, 259 participants were enrolled in this study.

## Attention network test

The attention network test (ANT) was used to assess the function of the three attention networks. Each trial began with a fixation cross being presented in the center of the screen. Cues were presented after a random interval of 400 to 1600 ms, as one of the four possible conditions: no cue, center cue, double cue, or spatial cue. In the center cue condition, fixation was replaced by an asterisk. In the double cue condition, the two asterisks were respectively displayed above and below the fixation. In the spatial condition, an asterisk was displayed either above or below the fixation. These asterisks served as cues to provide temporal information about the coming target stimuli, and remained visible for 100 ms. The spatial cue provided additional valid information about the location of target stimuli. The fixation cross was subsequently presented for 400 ms. Then, a target stimulus was presented in a certain position according to the previous cue. In the center cue and spatial cue conditions, target stimuli were presented in the same position as a cue. In the center cue and double cue conditions, target stimuli were presented in the center of the screen. Five target stimuli were arranged as horizontal arrows or lines that were severed as flankers under three different conditions: congruent condition (the other four arrows pointed in the same direction as the central arrow), incongruent condition (the other four arrows pointed in the opposite direction to the central arrow), or neutral condition (the four lines had no directional information). Target stimuli were maintained for 1,700 ms until the participant responded.

Participants were instructed to respond to the direction of the central arrow as fast and as accurately as possible. The participants pressed key "1" of a numeric keyboard with their left index fingers if the arrow pointed left, and key "3" with their right index fingers if the arrow pointed right. The numerical keyboard was aligned with the middle of the screen. Participants were required to concentrate on the fixation cross throughout the task.

ANT consisted of four blocks of 48 trials per person (192 in total). Within each block, all 48 trials were unique combinations of four cues condition (no cue, center cue, double cue, and spatial cue), three flanker conditions (congruent, incongruent, and neutral), two arrow directions (left or right), and two target display locations (above or below the fixation cross). ANT was presented and the data were recorded by Psychtoolbox (Fig. 1).

## Procedure

The experimenter informed the participants about the purpose of the experiment after they arrived at the laboratory. Then, the participants were instructed to sign the consent form and complete the IPAQ, MMSE, and exercise mode questionnaires. Edinburgh Handedness Inventory (*Oldfield, 1971*) and Waterloo Footedness Questionnaire Revised (WFQ-R; *Elias, Bryden & Bulmanfleming, 1998*) were also used to measure the laterality of the participants. Eligible participants started the ANT task after reading the ANT instructions by themselves. The ANT task was performed individually in a quiet and dimly lit room. All eligible participants performed a practice block consisting of 24 random trials.

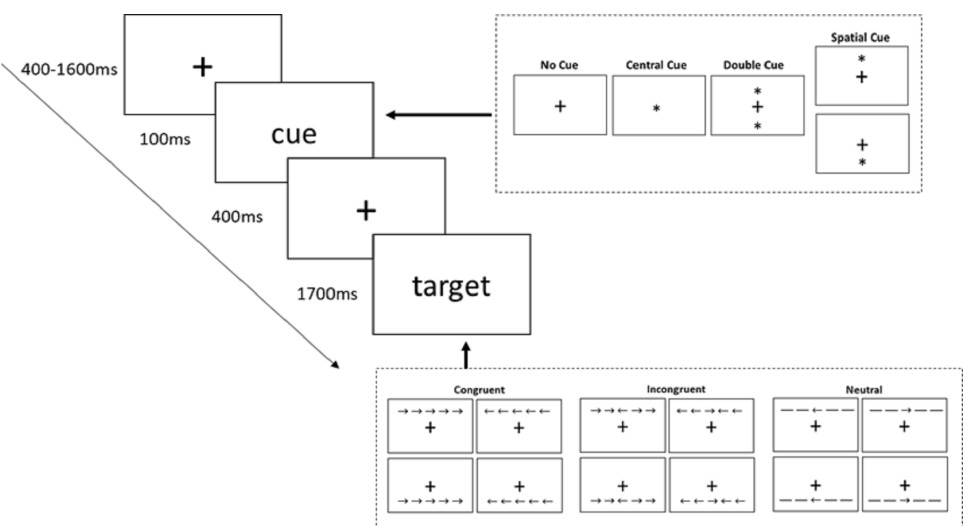

**Figure 1 Stimuli and experimental paradigm of the Attention Network Test (ANT).**

The test program counted their response accuracy (RA), and provided feedback at the end of the practice block. If the RA did not reach 80%, they had to perform another practice block. Participants with an RA exceeding 80% were allowed to perform the experimental blocks. There were four experimental blocks for each participant. Each experimental block consisted of 48 trials. When a block was completed, participants could take a break until they felt adequately rested. The next block was initiated by pressing any key. No feedback was provided during the experimental blocks. The whole ANT task took about 17 min to complete for each participant.

## Design and statistical analysis

The present study used a mixed factor design. The between-subjects variable was the exercise mode group. There were two within-subject variables: the cue type and flankers type. The dependent variables were reaction times (RTs) and accuracy rates. A group (3-open, closed, and control) ×cue type (4-no, central, double, and spatial) ×flanker type (3-congruent, incongruent, and neutral) mixed-design ANOVA was used to analysis the variables.

The efficiency of alerting was computed with RTs with no cue minus RTs with double cues, orienting was RTs with central cues minus RTs with spatial cues, and conflict resolution was RTs of incongruent flankers minus RTs of congruent flankers. The effect of different exercise modes on the three components of the attentional networks was evaluated using one-way ANOVA.

Receiver operating characteristic (ROC) analysis was conducted to evaluate whether the ANT components could discriminate older adults that used different exercise modes, and the area under the ROC curve (AUC) was calculated.

All statistical analysis were performed by SPSS 20.0. Significant levels were set at $p < 0.05$.

**Table 1  Main characteristics of the participants in the three groups.**

|  | Sedentary($n = 87$) | Closed-skill ($n = 87$) | Open-skill($n = 85$) |
|---|---|---|---|
| Male | 41 | 38 | 40 |
| Age (years) | $65.9 \pm 6.3$ | $65.5 \pm 5.8$ | $66.8 \pm 5.5$ |
| Education (years) | $11.4 \pm 2.3$ | $10.9 \pm 2.3$ | $11.3 \pm 2.6$ |
| Height (cm) | $161.3 \pm 5.8$ | $161.8 \pm 5.5$ | $163.0 \pm 7.1$ |
| Weight (kg) | $62.3 \pm 6.0$ | $62.1 \pm 5.9$ | $63.8 \pm 9.4$ |
| BMI (kg/m$^2$) | $24.0 \pm 2.3$ | $23.8 \pm 2.2$ | $24.0 \pm 2.9$ |
| MMES | $27.9 \pm 1.4$ | $28.2 \pm 1.6$ | $28.3 \pm 1.2$ |
| IPAQ (METs/week) | $4,458.8 \pm 2,770.8$ | $6,683.9 \pm 3,031.9$ | $6,737.4 \pm 2,631.0$ |

**Notes.**

MMSE, Mini Mental State Exam; IPAQ, International Physical Activity Questionnaire; METs, metabolic equivalents.

## RESULTS

### Demographic characteristics

No significant differences were found in age ($F_{(2,256)} = 1.10$, $p = 0.33$), education level (F $_{(2,256)} = 1.11$, $p = 0.33$), height ($F_{(2,256)} = 1.93$, $p = 0.15$), weight ($F_{(2,256)} = 1.47$, $p = 0.23$), BMI ($F_{(2,256)} = 0.21$, $p = 0.81$), and MMSE ($F_{(2,256)} = 1.82$, $p = 0.17$). However, a significant difference was observed for physical activity level ($F_{(2,256)} = 18.46$, $p < 0.01$). A LSD post-hoc comparison showed open- and closed- skill groups conducted a higher level of physical exercise compared to the sedentary group ($p < 0.01$), and that there were no significant difference between the open and closed-skill groups. The main characteristics of the participants are shown in Table 1.

### Mean reaction times

All incorrect trials or the trials that were three standard deviations from the individual mean were excluded from the RT analysis. A significant main effect of group (F$_{(2,256)} = 7.79$, $p < 0.01$, $\eta_p^2 = 0.06$) was obtained. Post hoc comparison showed that open- and closed-skill groups were faster compared to the sedentary group. A significant main effect was also observed for flanker type (F$_{(2,512)} = 1285.12$, $p < 0.01$, $\eta_p^2 = 0.83$) and cue type (F$_{(3,768)} = 188.50$, $p < 0.01$, $\eta_p^2 = 0.42$). RTs were longest under the no cue condition, and shortest under the spatial cue condition. RTs were longer under the incongruent condition compared to the congruent and neutral conditions. Significant interactions between flanker type and cue type (F$_{(6,1536)} = 18.16$, $p < 0.01$, $\eta_p^2 = 0.07$), and group and flanker type (F$_{(4,1024)} = 18.72$, $p < 0.01$, $\eta_p^2 = 0.13$) were obtained. The interaction analysis for flanker type and group showed that the open-skill exercise group was faster compared to the closed-skill exercise group and the sedentary group under incongruent conditions. In comparison, no differences were detected among the groups under the congruent and neutral conditions. The interaction analysis for cue and flanker type showed significant differences between the congruent and incongruent conditions, and the incongruent and neutral conditions, under all cue conditions. No significant differences between the congruent and neutral conditions were detected under any cue conditions. There was no significant interaction effect of group $\times$ cue type (F$_{(6,1536)} = 1.23$, $p = 0.29$, $\eta_p^2 = 0.01$), or group $\times$ cue type $\times$ flanker type (F$_{(12,3072)} = 1.01$, $p = 0.44$, $\eta_p^2 = 0.01$). The descriptive

**Table 2** Mean RTs (ms) and standard deviations of the three exercise mode groups according to cue and flanker type.

| Group | Flanker type | Cue type | | | |
|---|---|---|---|---|---|
| | | No cue | Center cue | Double cue | Spatial cue |
| | Congruent | 662.92 ± 97.65 | 638.88 ± 93.93 | 635.77 ± 84.52 | 629.05 ± 98.99 |
| Sedentary | Incongruent | 728.84 ± 106.75 | 730.01 ± 99.70 | 721.36 ± 97.90 | 701.28 ± 105.11 |
| | Neutral | 646.17 ± 99.03 | 625.88 ± 81.35 | 630.36 ± 93.30 | 622.26 ± 91.99 |
| | Congruent | 640.06 ± 68.25 | 614.99 ± 75.47 | 617.30 ± 73.66 | 605.30 ± 71.25 |
| Closed-skill | Incongruent | 698.26 ± 76.81 | 697.93 ± 80.24 | 687.14 ± 79.41 | 668.02 ± 77.45 |
| | Neutral | 625.32 ± 64.10 | 602.78 ± 63.19 | 605.63 ± 66.80 | 590.41 ± 57.21 |
| | Congruent | 622.63 ± 70.35 | 599.89 ± 73.85 | 596.86 ± 76.51 | 593.05 ± 74.06 |
| Open-skill | Incongruent | 665.28 ± 83.70 | 666.93 ± 73.94 | 654.36 ± 79.93 | 633.25 ± 77.97 |
| | Neutral | 615.44 ± 71.65 | 599.82 ± 61.84 | 590.92 ± 70.15 | 580.25 ± 70.97 |

data of the mean RTs and standard deviations of different exercise mode groups according to the cue and flanker type are shown in Table 2.

## Accuracy

Significant main effects of group ($F_{(2,256)} = 4.47$, $p = 0.01$, $\eta_p^2 = 0.03$) and flanker type ($F_{(2,512)} = 86.09$, $p < 0.01$, $\eta_p^2 = 0.25$) were obtained in the accuracy analysis. There was no significant main effect of cue type ($F_{(3,768)} = 0.30$, $p = 0.82$, $\eta_p^2 < 0.01$). There was no significant interaction between group and flanker type ($F_{(4,1024)} = 1.10$, $p = 0.36$, $\eta_p^2 = 0.01$), flanker type and cue type ($F_{(6,1536)} = 2.03$, $p = 0.11$, $\eta_p^2 = 0.01$), group × cue type ($F_{(6,1536)} = 2.38$, $p = 0.10$, $\eta_p^2 = 0.02$), or group × cue type × flanker type ($F_{(12,3072)} = 1.85$, $p = 0.21$, $\eta_p^2 = 0.01$). The descriptive data for the mean accuracy and standard deviations of the exercise mode groups according to cue and flanker type are shown in Table 3.

## Group differences on the efficiency of three attentional networks

One-way ANOVA was carried out on the efficiency of each component of the attentional system (alerting, orienting, and executive networks). A significant difference among the groups was obtained for the executive network ($F_{(2,256)} = 20.78$, $p < 0.01$). Post hoc analysis showed that open-skill exercisers reached a significantly higher executive network efficiency compared to closed-skill exercisers ($p < 0.01$) and sedentary controls ($p < 0.01$), and that closed-skill exercisers were significant higher than sedentary controls ($p < 0.01$). No significant differences were observed for alerting ($F_{(2,256)} = 0.64$, $p = 0.53$) and orienting ($F_{(2,256)} = 1.62$, $p = 0.20$) networks (Fig. 2).

Given that a marginally significant difference occurred in the overall reaction times among the groups ($F_{(2,256)} = 7.79$, $p < 0.01$, the RTs of the open-skill, closed-skill, and control groups were 618 ms, 638 ms, and 664 ms, respectively), the efficiency of the three attentional networks was divided by the overall RT of each participant. This efficiency is called the proportion scores of the attentional networks (*Gamboz, Zamarian & Cavallero, 2010*). There was a significant difference among the groups for the proportion scores in the executive network ($F_{(2,256)} = 15.99$, $p < 0.01$). Post hoc analysis showed that
**Table 3  Mean accuracy (%) and standard deviations of the exercise mode groups according to cue and flanker type.**

| Group | Flanker | Cue type | | | |
|---|---|---|---|---|---|
| | | No cue | Center cue | Double cue | Spatial cue |
| | Congruent | 97.41 ± 4.63 | 97.99 ± 5.99 | 96.26 ± 6.06 | 98.49 ± 3.56 |
| Sedentary | Incongruent | 97.13 ± 5.29 | 94.32 ± 6.71 | 93.97 ± 8.00 | 94.40 ± 6.32 |
| | Neutral | 97.77 ± 4.95 | 97.49 ± 4.81 | 97.99 ± 5.00 | 96.98 ± 4.66 |
| | Congruent | 96.77 ± 4.75 | 98.78 ± 3.28 | 97.99 ± 3.63 | 97.27 ± 4.74 |
| Closed-skill | Incongruent | 95.69 ± 7.88 | 94.25 ± 7.99 | 95.40 ± 6.57 | 94.25 ± 5.74 |
| | Neutral | 97.41 ± 5.09 | 97.92 ± 3.77 | 97.77 ± 4.15 | 98.20 ± 2.84 |
| | Congruent | 98.38 ± 3.22 | 97.94 ± 3.25 | 98.46 ± 3.18 | 98.46 ± 3.97 |
| Open-skill | Incongruent | 95.88 ± 4.48 | 96.91 ± 5.25 | 95.44 ± 6.35 | 96.91 ± 4.59 |
| | Neutral | 97.94 ± 2.95 | 96.99 ± 4.27 | 99.26 ± 2.24 | 98.46 ± 3.60 |

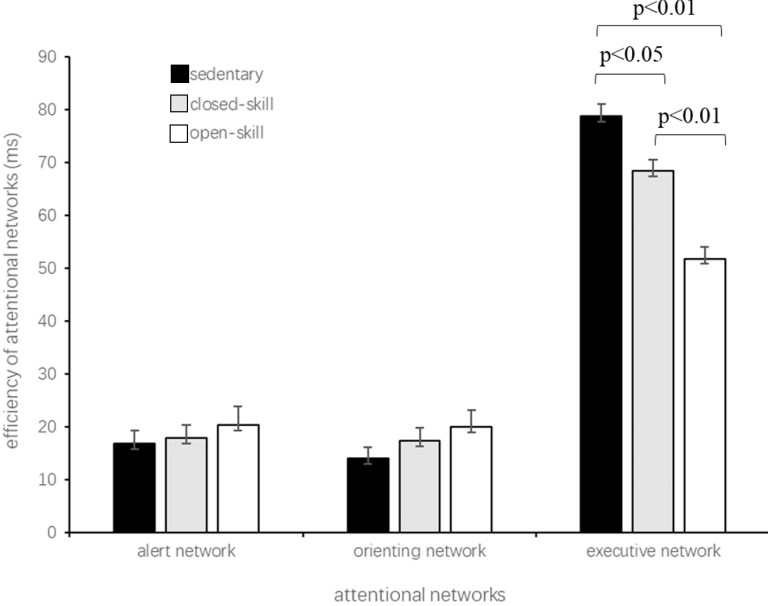

**Figure 2  Efficiency of the three attentional networks in older adults with different exercise modes.**

open-skill exercisers had a significantly higher proportion score compared to sedentary controls ($p < 0.01$). In comparison, no significant differences were observed for alerting ($F_{(2,256)} = 1.73$, $p = 0.18$) and orienting ($F_{(2,256)} = 2.22$, $p = 0.11$) networks (Fig. 3).

## Receiver operating characteristic (ROC) analysis

To evaluate whether the ANT components could discriminate older adults that used different exercise modes, receiver operating characteristic (ROC) analysis was conducted, and the area under the ROC curve (AUC) was calculated. Between the closed-skill group and the control group, the AUC value of the alerting network (AUC value = 0.46, $p = 0.41$, 95% CI [0.38–0.55]), orienting network (AUC value = 0.48, $p = 0.58$, 95% CI [0.39–0.56]), and

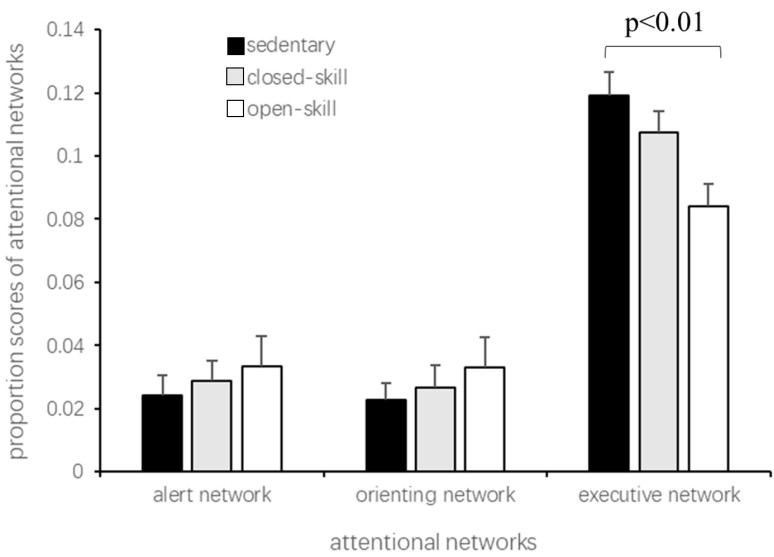

**Figure 3 Proportion scores of the three attentional networks in older adults with different exercise modes.**

executive network (AUC value = 0.58, $p = 0.09$, 95% CI [0.49–0.66]) was not significant. Between the open-skill group and the control group, the AUC value of the alerting network (AUC value = 0.43, $p = 0.13$, 95% CI [0.35–0.52]) and orienting network (AUC value = 0.44, $p = 0.20$, 95% CI [0.36–0.53]) was not significant. However, the AUC value of the executive network (AUC value = 0.77, $p < 0.01$, 95% CI [0.70–0.84]) was significant. Between the closed-skill group and the open-skill group, the AUC value of alerting network (AUC value = 0.46, $p = 0.38$, 95% CI [0.37–0.55]) and orienting network (AUC value = 0.47, $p = 0.46$, 95% CI [0.38–0.55]) was not significant. However, the AUC value of the executive network (AUC value = 0.69, $p < 0.01$, 95% CI [0.61–0.77]) was significant (Fig. 4). According to *Hosmer Jr, Lemeshow & Sturdivant (2013)*, AUC > 0.7 represents a good discrimination criterion (*Hosmer Jr, Lemeshow & Sturdivant, 2013*).

## DISCUSSION

The current study aimed to evaluate the relationship between different exercise modes and the attentional networks of older adults (>60 years old). Two-hundred and fifty-nine older adults were allocated in the open-skill group, closed-skill group, or sedentary group based on the exercise mode questionnaire. Group differences on the executive network were found in the ANOVA analysis. The interaction showed that the open-skill exercisers were faster under the incongruent conditions compared to closed-skill exercisers and sedentary controls. Of note, reduced executive control was accompanied by slower processing speeds; consequently, it was possible to calculate the proportion scores of the attentional networks. After controlling for general processing speed, open-skill exercisers exhibited better executive control compared to non-exercisers. Alerting and orienting networks were similar among the groups, which was consistent with a previous study showing that
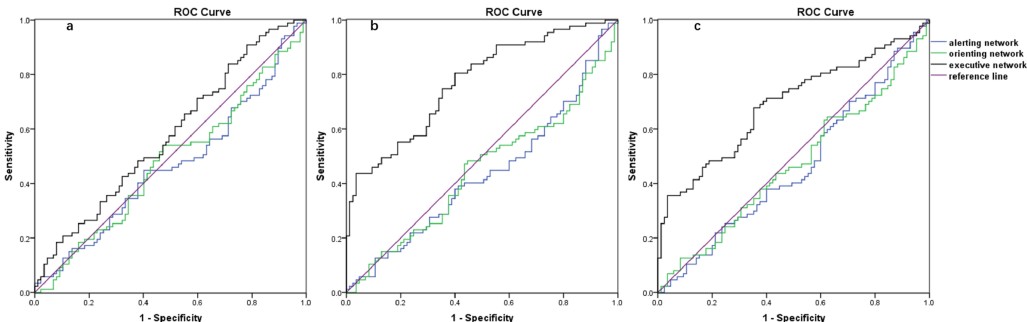

**Figure 4** **ROC curves for the three attentional networks.** (A) The closed-skill group and the control group; (B) the open-skill group and the control group; (C) the closed-skill group and the open-skill group.

young adult chronic exercisers exhibit equivalent levels of alerting and orienting to their sedentary controls (*Pérez et al., 2014*). To strengthen the effect of selective enhancement, ROC analysis showed that the executive network had a larger power than the alerting and orienting network at discriminating between open-skill exercisers and the controls in older adults. These results are similar with a previous study showing that college-level table tennis athletes exhibit selectively enhanced executive control of the attentional networks (*Wang, Guo & Zhou, 2016*). Thus, the selective enhancement of the executive control network of open-skill exercise occurs in both younger and older adults.

Aging is associated with the impairment of attention (*Verhaeghen & Cerella, 2002*). Studies that focused on the relationship between aging and the three attentional networks showed that alerting, orienting, and executive function are differentially affected by aging. In general, these studies concluded that the orienting network remains intact with aging (*Fernandez-Duque & Black, 2006*; *Gamboz, Zamarian & Cavallero, 2010*; *Jennings et al., 2007*; *Williams et al., 2016*). However, the results obtained for the other two networks are less consistent. For example, studies focusing on the age differences of ANT showed an age-related decline for the executive network (*Gamboz, Zamarian & Cavallero, 2010*; *Jennings et al., 2007*); however, when age-related slowing of reaction time was accounted for, there were no differences between younger and older individuals (*Fernandez-Duque & Black, 2006*; *Williams et al., 2016*). Most studies on the alerting network showed an age-related decline, with just one study reporting no age-related decline (*Fernandez-Duque & Black, 2006*). More specifically, one study focusing on the efficiency of attentional components in elderly people with mild neurocognitive disorders (NCD) showed that the NCD group was less efficient with respect to the executive control network, and had a slower processing speed compared to healthy elders (*Lu et al., 2016*).

Previous studies showed that attentional function is enhanced by physical exercise (*Hawkins, Kramer & Capaldi, 1992*; *Sanabria et al., 2011*; *Tsai et al., 2016*) and attention training (*Tang & Posner, 2009*). To the best of our knowledge, our study is the first to investigate the relationship between exercise mode and the attentional networks in older adults. Physical exercise was divided into two modes in the present study: open-skill

exercises and closed-skill exercises. Compared to closed skill exercises, open-skill exercises are performed in unpredictable environments that require a higher investment of cognitive effort. Open-skill exercises combine physical exercise and cognitive training simultaneously (*McMorris, 2014*). Because of their cognitively challenging environments (*Fabel et al., 2009*), open-skill exercises are more effective at inducing cognitive benefits for athletes (*Voss et al., 2010*; *Wang et al., 2013*), older adults (*Dai et al., 2013*; *Huang et al., 2014*), and developmental coordination disorder (DCD) children (*Tsai, 2009*). The results of the current study supported those focused on older adults, with open-skill exercisers exhibiting better executive function (*Dai et al., 2013*; *Huang et al., 2014*), visuospatial attention (*Tsai et al., 2016*), and working memory (*Guo et al., 2016*) compared to closed-skill exercisers.

More interestingly, no significant differences were observed between closed-skill exercisers and sedentary older adults on the proportion scores of the attentional networks in the present study. Proportion scores were used to control for the possibility that group differences in reaction time affected the efficiency of alerting, orienting, and executive control networks. The results of proportion score analyses mirrored previous studies among older adults, showing no behavioral differences between closed-skill exercisers and sedentary adults for the switch task (*Dai et al., 2013*) and visuospatial short-term memory (*Guo et al., 2016*). In comparison, significant differences were observed between open-skill exercisers and sedentary controls. This difference might be explained by the fact that these previous studies used cross-sectional designs, with a relatively small number of participants. Compared to closed-skill exercise, open-skill exercises enhanced the cognitive function of older adults in the current study.

The present study had some limitations. From the perspective of experimental design, the cross-sectional design only revealed a possible relationship between the exercise mode and the attentional networks in older adults. Also, open-skill exercises require more social interaction than closed-skill exercises, which might also explain the differences between the groups. Thus, longitudinal studies are needed to reveal these causal relationship in the future. From the perspective of experimental implementation, the testing time was not strictly controlled. Knight (2013) found that older adults performed best on the alerting network in the mornings, while the other two networks were not affected by time of day (*Knight & Mather, 2013*). The present study was performed in the morning or in the afternoon; thus, the non-significant difference in the alerting network might be masked by the timing of the tests.

## CONCLUSIONS

The current study provided further evidence on the association of physical exercise with better attentional networks in older adults. Compared to closed-skill exercisers and sedentary controls, open-skill exercisers exhibited a selective enhancement of the executive control network, whereas no differences were observed in the orienting and alerting networks across the three groups. The important public health implications of our results are that open-skill exercises (which combine physical exercise and cognitive training) might provide a potentially effective type of intervention for older adults to maintain or enhance attentional function.

### Funding

This work was supported by the Natural Science Foundation of Jiangsu Province (BK20180926) and the Natural Science Foundation of Higher Education of Jiangsu Province (18KJD190004). The funders had no role in study design, data collection and analysis, decision to publish, or preparation of the manuscript.

### Grant Disclosures

The following grant information was disclosed by the authors:
Natural Science Foundation of Jiangsu Province: BK20180926.
Natural Science Foundation of Higher Education of Jiangsu Province: 18KJD190004.

### Competing Interests

The authors declare there are no competing interests.

### Author Contributions

- Biye Wang conceived and designed the experiments, authored or reviewed drafts of the paper, and approved the final draft.
- Wei Guo analyzed the data, performed the experiments, prepared figures and/or tables, and approved the final draft.

### Human Ethics

The following information was supplied relating to ethical approvals (i.e., approving body and any reference numbers):

This study was approved by the Ethics Committee of the Shanghai University of Sport (No. 2017044).

### Data Availability

Data is available in a Supplemental File.

### Supplemental Information

Supplemental information for this article can be found online at http://dx.doi.org/10.7717/peerj.8364#supplemental-information.

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
