# Peer review of "Exercise mode and attentional networks in older adults: a cross-sectional study"

_PeerJ, doi:10.7717/peerj.8364_

## Round 0.1 · original submission · Minor Revisions

The three reviewers and I see many positives from the study described in this manuscript, but also highlight some issues for you to consider. In relation to reviewers to's comment regarding statistical power and type I error, I would recommend you take on board their comment around calculating the statistical power of your study. However, in relation to the point that this relative lack of power may underpin the statistically significant findings is questionable, as a lack of statistical power typically results in a type II error, in which you are more likely to find no statistically significant findings.

·

Basic reporting

-The English language is clear. However, there are some tiny mistakes. For example, in line 180 you used "trails" instead of "trials". Maybe I am missing something ( I am not an English native speaker).

Experimental design

-Why do you choose the International Physical Activity Questionnaire and an exercise mode questionnaire?. Previous research using the same questionnaire/s should be included as well as reliability and statistical validity.

-Sample size should be justified by providing a power statement.

-How do you measure the laterality of the participants?. Edinburgh Handedness Inventory should be included (Oldfield, 1971). Even more, Waterloo Footedness Questionnaire Revised (WFQ-R; Elias, et al., 1998), would be desirable to include in the described methods.

- Was the numerical keyboard aligned with the middle of the screen?. Please, include this information.

Validity of the findings

The main result shows that that open-skill group had significantly higher executive network efficiency compared to the closed-skill and sedentary groups. The major implication (or conclusion) is well established, highlighting that open-skills exercises improve executive control.


Overall, data are new and statistically well controlled. and important for the field.

·

Basic reporting

No comment

Experimental design

No comment

Validity of the findings

According to the results of various studies and meta-analyses (see references bellow), the effect of physical exercise on executive function is very small (<0.2), which means that the sample size of this study is too small to detect such an effect. Authors should have conducted a power analysis based on reported effect sizes before selecting their sample size. Consequently, the reported "significant" findings are most probably due to type I error.

Also, please note that in a recent meta-analysis (Sanabria, D., et al. (2019). The relationship between vigilance capacity and physical exercise: a mixed-effects multistudy analysis. PeerJ, 7, e7118.) we find an effect of externally-paced sports (open-skill) on alertness, measured by the Psychomotor Vigilance Task. The fact that this study does not detect this much larger effect of exercise on alertness is further indicative of its low statistical power, although it could also be related to the sensitivity of the ANT.

References
Xue, Y., Yang, Y., & Huang, T. (2019). Effects of chronic exercise interventions on executive function among children and adolescents: A systematic review with meta-analysis. British Journal of Sports Medicine, bjsports-2018-099825. https://doi.org/10.1136/bjsports-2018-099825
Northey, J. M., Cherbuin, N., Pumpa, K. L., Smee, D. J., & Rattray, B. (2018). Exercise interventions for cognitive function in adults older than 50: A systematic review with meta-analysis. British Journal of Sports Medicine, 52(3), 154-160. https://doi.org/10.1136/bjsports-2016-096587
Verburgh, L., Königs, M., Scherder, E. J. A., & Oosterlaan, J. (2013). Physical exercise and executive functions in preadolescent children, adolescents and young adults: A meta-analysis. British Journal of Sports Medicine. https://doi.org/10.1136/bjsports-2012-091441
Fedewa, A. L., & Ahn, S. (2011). The effects of physical activity and physical fitness on children’s achievement and cognitive outcomes: A meta-analysis. Research Quarterly for Exercise and Sport, 82(3), 521-535. https://doi.org/10.1080/02701367.2011.10599785
Etnier, J. l., Salazar, W., Landers, D. m., Petruzzello, S. j., Han, M., & Nowell, P. (1997). The influence of physical fitness and exercise upon cognitive functioning: A meta-analysis. / L ’ influence de la condition physique et de l ’ exercice sur les fonctions cognitives: une meta analyse. Journal of Sport & Exercise Psychology, 19(3), 249-277.
Young, J., Angevaren, M., Rusted, J., & Tabet, N. (2015). Aerobic exercise to improve cognitive function in older people without known cognitive impairment. En Cochrane Database of Systematic Reviews. https://doi.org/10.1002/14651858.CD005381.pub4
Angevaren, M., Aufdemkampe, G., Verhaar, H. J. J., Aleman, A., & Vanhees, L. (2008). Physical activity and enhanced fitness to improve cognitive function in older people without known cognitive impairment. The Cochrane Database of Systematic Reviews, (3), CD005381. https://doi.org/10.1002/14651858.CD005381.pub3
Singh, A. S., Saliasi, E., van den Berg, V., Uijtdewilligen, L., de Groot, R. H. M., Jolles, J., Chinapaw, M. J. M. (2019). Effects of physical activity interventions on cognitive and academic performance in children and adolescents: A novel combination of a systematic review and recommendations from an expert panel. British Journal of Sports Medicine, 53(10), 640-647. https://doi.org/10.1136/bjsports-2017-098136
Kramer, A. F., & Colcombe, S. (2018). Fitness Effects on the Cognitive Function of Older Adults: A Meta-Analytic Study-Revisited. Perspectives on Psychological Science: A Journal of the Association for Psychological Science, 13(2), 213-217. https://doi.org/10.1177/1745691617707316

Additional comments

I would recommend that the authors run a power analysis based on the size of the reported effects of exercise on executive function and increase the sample size accordingly.

·

Basic reporting

The manuscript is well written but requires proofreading, e.g. in the abstract in the Background section 'the attentional network' is ambiguous as there are three types of attention and related networks are rightly stated in the Introduction.
The recent literature is referred to, there is another, older, review on RCTs which could be of use (https://www.ncbi.nlm.nih.gov/pubmed/24862109);
The structure of the manuscript is good and the results are commented in relation to the hypotheses.
Raw data need clearer headings or a legend tab.

Experimental design

The research question is clear and the paradigm used is standard for the question explored. The utilization of 'natural' groups has limitations, but the study is still sound.

Validity of the findings

The analyses conducted are in line with other papers utilising the ANT and the conclusions are supported. One limitation that should be acknowledged is the likely different level of social interaction in the two kinds of exercise groups, open skill does not only require to respond to an unpredictable environment, but, often, requires some form of social interaction. Consider ANCOVA for amount of regular exercise, or correlation with results (although the sample is not very large).

Additional comments

The study is interesting and the manuscript is well written.

---

## Round 0.2 · Minor Revisions

The two reviewers and I appreciate the hard work you've made in attending to the initial constructive criticism of this manuscript. The manuscript is now considered to be scientifically sound for publication, but we still have some concerns regarding the level of written English in parts of the manuscript. Can you please ensure this manuscript is critically proofread before the final submission to ensure the standard of English is high. If required, please seek the assistance of a professional proofreader in this regard.

·

Basic reporting

No comment.

Experimental design

No comment.

Validity of the findings

No comment.

Additional comments

All my remarks have been addressed in a satisfactory manner.

·

Basic reporting

As this is the second round of review please find my final comments below

Experimental design

Good

Validity of the findings

Good

Additional comments

The authors have increased the sample size and made some appropriate changes to the manuscript, there is proofreading still needed; the study is interesting and I feel that once the final proofreading has occurred it is ready for publication.

---

## Round 0.3 · Minor Revisions

Thanks for accessing an English-speaking proofreader as it has substantially improved the quality of the written English in this revised manuscript. A few minor edits that would further improve the readability of the manuscript include the following points that need to be addressed before this paper can be accepted for publication.
Line 92: please change this phrase to “that older adults who participate…”.
Line 228 – 230: I am unclear what questionnaires you are referring to here on line 228 and how this than reflects the Chinese version of the IPAQ. Would it be better to separate this into two sentences and explicitly describe what the “two questionnaires” on line 228 actually are?
Line 238 – 239: please provide a reference to the data used in this power analysis.
Line 306 – 314: please indicate the level of statistical significance used in the one-way ANOVA and what type of post-hoc test was performed.
Line 447 – 462: you report data here for a ROC analysis, but do not mention this within the statistical analysis section. Please provide sufficient detail on how this analysis was performed in an expanded statistical analysis section.
Line 540: should just be written as “Proportion scores were used to control for the possibility that group differences in…”.

---

## Round 0.4 · accepted · Accept

Thank you very much for your attention to improving this manuscript. I am now happy to recommend it be accepted for publication in PeerJ.